# Expression of the *CpXTH6* and *CpXTH23* Genes in *Carica papaya* Fruits

**DOI:** 10.3390/ijms26104490

**Published:** 2025-05-08

**Authors:** Melvin E. Zúñiga-Hernández, Raymundo Rosas-Quijano, Miguel Salvador-Figueroa, Alfredo Vázquez-Ovando, Didiana Gálvez-López

**Affiliations:** Instituto de Biociencias, Universidad Autónoma de Chiapas, Boulevard Príncipe Akishino Sin Número, Colonia Solidaridad 2000, Tapachula 30789, Chiapas, Mexico; melvin.zuniga9508@gmail.com (M.E.Z.-H.); raymundo.rosas@unach.mx (R.R.-Q.); miguel.salvador@unach.mx (M.S.-F.); jose.vazquez@unach.mx (A.V.-O.)

**Keywords:** xyloglucan, fruit development, cell wall, qPCR, papaya

## Abstract

Mexico is the center of origin and the leading exporter of papaya (*Carica papaya*) to the United States of America and Canada. The changes in the fruit’s firmness during ripening result from the action of several enzymes implicated in the synthesis/hydrolysis of cell wall polysaccharides. A vast family of genes encodes xyloglucan endotransglucosylase/hydrolase (XTH) enzymes, which act on cellulose-bound xyloglucan bonds. There are few reports on the action of the *XTH6* and *XTH23* genes; therefore, their participation in the fruit development and maturity processes has yet to be fully known. The expression levels of the *CpXTH6* and *CpXTH23* genes, and their correlation with firmness, at different stages of development and ripening of the *C. papaya* fruit were determined in this work. The *CpXTH6* and *CpXTH23* genes reached their highest expression level during fruit development. These results suggest that these genes are activated in papaya mainly during fruit development to encode the enzymes that allow cell growth and maintain fruit firmness. These findings could be used to target papaya breeding texture quality and the speed of fruit growth.

## 1. Introduction

Mexico is the center of origin of papaya, and many families depend on its production. In 2023, Mexico produced approximately 1,135,000 tons of papaya, securing fifth place among the world’s top producers of the fruit. Despite not leading in production volume, Mexico continues to dominate the global export market. With international sales reaching USD 123.7 million, the country holds the position of the world’s leading exporter of papaya, underscoring its efficiency and competitiveness in international trade [1]. The texture of the fruit is an important quality criterion that influences consumer acceptance, a determining factor in the postharvest process of the fruit. Cell walls make a significant contribution to the texture of the fruit. As in other fleshy fruits, the primary cell walls of papaya consist of a cellulose framework embedded in a matrix of hemicelluloses, pectins, and structural proteins [2,3,4]. The most abundant hemicellulose in the cell wall is xyloglucan; this long polysaccharide can form hydrogen bonds with cellulose and pectin microfibrils, providing stability and firmness to the cell wall [4,5]. Cellular elongation and expansion in fruits is a complex but well-coordinated process which is carried out through the action of various enzymes that participate in the formation and union of glycosidic bonds, thus strengthening the rigidity of the cell wall [6,7].

Meanwhile, the fruit softens during the ripening due to the disassembly of union chains between cell wall polysaccharides [3,4,5,6]. This pulp softening likely occurs through the reduction of cell-to-cell adhesion resulting from the dissolution of the polysaccharides of the primary cell wall and the middle lamella caused by the action of hydrolases [7]. Various enzymes hydrolyze cellulose, hemicellulose, and pectin molecules [6,8], including cellulases, glycosidases, β-galactosidase, α-arabinofuranosidase, polygalacturonase, pectin methyl esterases, endoglucanases, transglycosidases, esterases, xyloglucan endo/hydrolases (XEH), xyloglucan endotransglucosilases (XETs), and xyloglucan endotransglucosylases/hydrolases (XTHs) [7,9,10,11,12].

The XTH family of enzymes acts on xyloglucan bonds bound to cellulose [9]. Some members of this family (e.g., XET) have been found to act during fruit development [13], while others (e.g., XEH) act predominantly in the ripening stage of the fruit [10], while XTH acts in both [8,14].

The XTH family, encoded by various genes, is grouped into several subfamilies. The expression of *XTH* genes has been reported during the development and maturation stages in different fleshy fruits such as *SlXTH1*, *SlXTH2*, *SlXTH3*, *SlXTH5*, *SlXTH6*, *SlXTH7*, *SIXTH10*, *SlXTH12*, and *SlXTH23* in tomato (*Solanum lycopersicum*) [8,9,15]; *FcXTH1*, *FcXTH2*, *FvXTH6*, and *FvXTH9* in strawberry (*Fragaria chiloensis*) [16,17]; *AdXTH5*, *AdXTH6*, and *AdXTH10* in kiwi (*Actinidia deliciosa*) [10]; *MdXTH2* and *MdXTH6* in apple (*Malus domestica*) [9,10,11,18]; *CmXTH1* and *CmXTH3* in melon (*Cucumis melo*) [19]; *PavXTH14* and *PavXTH15* in sweet cherry [20]; and *DkXTH1*, *DkXTH2*, *DkXTH4*, and *DkXTH5* in persimmon (*Diospyrus persimmon*) [14].

Determining the expression of the *XTH* gene family could be a critical factor in the delay of the softening of the fruit. The expression of the *XTH6* and *XTH23* genes has been analyzed predominantly during the ripening of the fruits, and few works have reported on the expression during the development of organs. Atkinson et al. (2009) and Muñoz et al. (2013) [9,10] have described the correlation between the decrease in the firmness of fleshy fruits and the increase in the expression of *XTH6* genes in apple, kiwi, and tomato fruits at the ripening stage, while Muñoz et al. (2013) [9] reported that the *XTH23* gene shows low expression during the tomato maturity stage. This type of analysis allows inferring the relationship between the firmness of the fruits and the expression of the *XTH* genes, which is the first step in finding genes related to the softening of fleshy fruits.

Meanwhile, the expression of these genes in papaya fruits has neither been reported nor associated with softening. In a recent study about *C. papaya* carried out by this research group, the *CpXTH2* and *CpXTH5* genes were identified in *C. papaya* [21]. In that work, the highest expression of the *CpXTH2* and *CpXTH5* genes was during the development of the fruit. Therefore, this work aimed to determine the expression levels of the *CpXTH6* and *CpXTH23* genes in *C. papaya* at different stages of fruit development and ripening.

## 2. Results

### 2.1. Expression of CpXTH6 and CpXTH23 Genes in C. papaya Fruits

The expression patterns of the *CpXTH6* and *CpXTH23* genes during the development and organoleptic maturation of papaya fruit are depicted in Figure 1 and Figure 2. Both genes were expressed during both stages of the fruit. At the fruit development stage, the expression of the *CpXTH6* gene (Figure 1) increased 16.4 times from the beginning of fruit development (S1D) until the fruit stopped growing (S8D). On the other hand, during the transition stage between physiological ripeness (S8D) and the beginning of organoleptic ripeness (SGM) of the fruit, the expression of the *CpXTH6* gene (Figure 1) decreased 3.6 times, a trend that continued until it reached the S6M phase, where the expression of this gene was 11.1 times lower when compared to the SGM phase (Figure 1).

Contrary to the expression dynamics of the *CpXTH6* gene, during fruit development, the expression of the *CpXTH23* gene was always downward; that is, the maximum expression was found in the S1D stage, decreasing 10.9 times upon reaching the S8D stage (Figure 2). During the transition stage between S8D and SGM, the expression of the *CpXTH23* gene increased 3.3 times and then resumed the downward pattern until reaching the S6M stage, where the expression was 5.2 times lower than that observed in SGM (Figure 2).

### 2.2. Firmness of C. papaya Fruits

The firmness values of papaya fruit during the different stages of development and organoleptic ripeness are shown in Figure 3. The firmness values were statistically similar during the transition from stages S1D to S6D (average 12.8 N). However, in the transition from S1D to S8D, the firmness decreased by 23.54% and was statistically significant (*p* < 0.05). This decline is likely due to S8D being the final stage of development, where fruit softening begins to accelerate. On the other hand, a significant trend was the more accelerated decrease in firmness during the organoleptic ripeness stage, reaching, in stage S6M, values that represented only 3.1% of the average value of stages S1D–S6D (Figure 3), reflecting the rapid breakdown of cell wall components characteristic of this ripening phase [3,4].

### 2.3. Correlation Between Relative Expression of CpXTH Genes and Fruit Firmness

Figure 4 illustrates the significant correlation between the relative expression of the *CpXTH6* and *CpXTH23* genes and fruit firmness. This correlation is particularly noteworthy during the stages of fruit development and organoleptic ripening. The *CpXTH23* gene consistently shows a positive correlation with fruit firmness (Figure 4B,D), while the *CpXTH6* gene, with its mixed behavior, demonstrates the complexity of this relationship, showing a negative correlation during development and a positive correlation during organoleptic ripening (Figure 4A,C).

## 3. Discussion

The results of the expression of the *CpXTH6* gene in the present study show that it is expressed mostly during the later phases (S6D and S8D) of the development stage of the fruit, which matches the findings of Miedes and Lorences (2009), [8] who reported that the *SlXTH6* homologue has a higher expression during the stages of tomato development and likewise correlated gene expression with the phenomenon of cell elongation through XET activity. The *MdXTH6*, *AdXTH6*, and *SIXTH6* genes were expressed at low levels during the ripening of apple, kiwi, and tomato, respectively [9,10]. Additionally, Zhang et al. (2017) [22] found that the *XTH6* gene exhibited two expression peaks, one in the early and another in the late stages of Taishanzaoxia apple fruit development, suggesting it plays a role in both phases. Likewise, Li et al. (2019) [23] showed that *Ac(MD2)XTH6* has normal transcript levels during the early immature stages, a high transcript level in the final immature stage, and a low transcript level during the fruit ripening stage. Given the double function of the XTH enzymes (glycosylation or hydrolysis) during the remodeling of the fruit cell wall, the higher levels of transcripts found in the development process of *C. papaya* could indicate that this gene codes for the enzymes that contribute to the expansion of the fruit and to the modification of the structure of the cell wall xyloglucan, as reported by Witsari et al. (2019) [17] for *FvXHT6*. As suggested by Vissenberg et al. (2005) [24], the XET action on the xyloglucan–cellulose bonds could regulate the extensibility of the cell wall and plant growth; therefore, it is possible that the *CpXTH6* gene product has transglucosylase activity. In addition, the drastic drop in the expression of this gene suggests that a decrease in its expression is required to start the softening of the fruit. However, other *XTH* members not analyzed may also be involved in these processes, and further genetic analyses are needed to fully validate these findings.

Additionally, the expression of the *CpXTH23* gene was found to record its maximum value during the S1D development stage of the fruit. As far as the authors of this work are aware, the expression of this gene in other fleshy fruits during development has not been reported before. The high expression of this gene in early stages of fruit development suggests that this gene is probably activated from the start of the anthesis process or just after the pollination and fertilization process that leads to fruit development. This phenomenon was observed by Han-Ye et al. (2016) [13], who found an expression of the *DkXTH6* and *DkXTH7* genes during the anthesis process in persimmon. This finding suggests that the expression of the *CpXTH23* gene is related to the cell elongation process in the early stages of fruit development, as it encodes enzymes that participate in the rapid division and expansion of cells, when an activity of the enzymatic product is more likely to take place. Therefore, XET shows transglycosylase activity and, consequently, maintains rigidity.

The *CpXTH23* gene had a lower transcription level during fruit ripening; however, a significant increase in expression was observed in the SGM stage with respect to the last stage of development. Studies carried out in ripe tomato fruits showed that the *SlXTH23* gene has a low transcription level [9]. This expression pattern has also been observed with *SlXTH1* in transgenic tomatoes, whose enzyme products have greater transglycosylation activity; this activity could be responsible for the structural changes of the cell wall and, therefore, can alter its extensibility [7]. These results suggest that the expression of the *XTH23* genes is not strongly involved in the softening of these fruits.

For their part, the firmness results only show a significant difference between the first (S1D) and the last (S8D) stages of the development process, as this stage is not characterized by the loss of firmness. However, there is a significant decrease (*p* < 0.05) between the stages analyzed (SGM, S2M, and S4M) in the maturation stage. Santamaria et al. (2009) [25] described a similar loss of firmness in ripe fruits of *C. papaya*.

On the one hand, the negative correlation between the *CpXTH6* gene expression and the firmness during the developmental stage (Figure 4A) is explained by the progressive increase in the level of transcription and the minimal decrease in firmness. The positive correlation between the expression of the *CpXTH6* gene and the firmness during the ripening stage (Figure 4C) can be explained by the decrease in the transcriptional activity of the *CpXTH6* gene and the firmness of the fruit. This finding may mean that this gene plays a non-essential role in the depolymerization of the wall during the last stages of maturation. These results match the findings of Atkinson et al. (2009) [10] during the ripening of kiwi fruits, when the expression of the *AdXTH6* gene had a positive correlation with firmness.

On the other hand, the positive correlation in the expression of the *CpXTH23* gene with the firmness during both the development and maturation stages (Figure 4B,D) is explained by the progressive decrease of the two variables. This phenomenon may mean that the activity of this gene is also not essential for the depolymerization of the wall and is therefore related to the maintenance of the integrity of the cell wall and the firmness of the fruit. This same correlation behavior has been previously reported between the expression of *XTH* genes and the firmness at ripening of persimmons [14], peaches [26], and sweet cherries [20].

Therefore, the role of the *CpXTH6* and *CpXTH23* genes, better expressed during fruit development, could be related to the maintenance of the structural integrity of the cell wall; meanwhile, the decrease in the expression of these genes during ripening would lead to the softening of the fruit.

## 4. Materials and Methods

### 4.1. Biological Material

Papaya (*C. papaya*) fruits of the Maradol variety were obtained from the AYOL Agroecological Farm, located in Tapachula, Chiapas, Mexico (14°49′45.45″ N, 92°17′45.54″ W, 25 m.a.s.l). Fruits in various stages of development (SD) and organoleptic maturity (SM) were harvested and classified according to Aguilar-Velázquez et al. (2021) [21] and Santamaría-Basulto et al. (2009) [25]. For the SD, the fruits were analyzed 15 days after anthesis (DAA) or stage 1 (SD1), 60 DAA or SD4, 90 DAA or SD6, and 110 DAA or SD8; lengths of 6, 14, 17, and 20 cm were recorded, respectively (Figure 5). For SM, the fruits were chosen at the SGM, S2M, S4M, and S6M stages (Figure 6). The fruits were transported to the laboratory, where they were washed with distilled water; the mesocarp was separated, cut into cubes of approximately 0.5 cm^3^, and stored at −80 °C until use.

### 4.2. Total RNA Extraction

Total RNA extraction was performed using the TRIZOL^®^ (Invitrogen™, Carlsbad, CA, USA) method and purified with the RNeasy Plant Mini Kit (QIAGEN^®^, Hilden, Germany) after maceration of the mesocarpic tissue of the fruits with liquid nitrogen supplemented with 1 g of polyvinylpyrrolidone (Sigma-Aldrich^®^, St Louis, MO, USA). The RNA obtained was quantified in a Genova Nano spectrophotometer (JENWAY^®^, Staffordshire, United Kingdom), and the quality was verified in a 0.8% agarose gel (Bio-Rad Laboratories, Inc., Berkeley, CA, USA).

### 4.3. cDNA Synthesis

The cDNA synthesis was performed with the High-Capacity Reverse Transcription Kit (Applied Biosystems^®^, Waltham, MA, USA), using RNA adjusted to 40 ng/µL.

### 4.4. Oligonucleotides and Probes Used

The sequences of the oligonucleotides and their probes of the *XTH6*, *XTH23*, and *actin* genes were selected from the information of the accession FJ696416.1 of *C. papaya* deposited in 2018 at the NCBI (Table 1) using the software Primers Express^®^ version 3.0.1 (Applied Biosystems^®^, Waltham, MA, USA). The probes for the *CpXTH6* and *CpXTH23* genes were labeled with the FAM dye, while the actin gene probe was labeled with the VIC dye (TaqMan^®^ Probes, Applied Biosystems^®^, Waltham, MA, USA).

### 4.5. Gene Expression Analysis by qPCR

The expression of the *CpXTH6*, *CpXTH23*, and actin genes in each of the SD and SM of *C. papaya* fruits was quantified by qPCR (Applied Biosystems^®^ StepOnePlus™ real-time PCR, Waltham, MA, USA), following the TaqMan^®^ Gene protocol Expression Assays from the same trading company. The PCR mix was prepared as follows: 10 µL of 2X TaqMan^®^ Gene Expression Master Mix, 1 µL of TaqMan^®^ Probe (10 µM), 1 µL of Primer Forward (10 µM), 1 µL of Primer Reverse (10 µM), 5 µL RNase-free water, and 2 µL cDNA. The mix was plated in 96-well optical plates (StepOnePlus™) with two negative controls consisting of 18 µL of the PCR mix without cDNA and 2 µL of sterile Milli-Q water. The qPCR program used was an initial stage of 2 min at 50 °C and 15 s at 95 °C by 1 min at 60 °C for 40 cycles. The relative expression (relative expression = 2^−ΔΔCT^) of the *CpXTH6* and *CpXTH23* genes was determined with the equation: ΔΔCt = ΔCt (gene of interest) − ΔCt (calibrator gene) (Livak and Schmittgen, 2001) [27]. All reactions were performed three times.

### 4.6. Instrumental Firmness of C. papaya Fruits

The firmness of the mesocarp of *C. papaya* fruits in the different SD and SM, expressed in Newtons (N), was determined using a texturometer (LLOYD Instruments Plus^®^, Hampshire, United Kingdom) in penetrometer mode, based on the method reported by Santamaría- Basulto et al. (2009) [25]. Ten cubes (ten replicates) of 1.5 cm edge mesocarp were used for each SD and SM. The penetrometer was operated at a speed of 100 mm s^−1^, a maximum depth of 100 mm, and a maximum duration of 60 s per determination.

### 4.7. Data Analysis

All of the gene expression data for *CpXTH6* and *CpXTH23* (2^−ΔΔCT^ values), as well as the instrumental firmness measures, were subjected to an analysis of variance and afterward to a means separation analysis by Tukey’s test (*p* < 0.05). In addition, a Pearson conversion analysis was carried out between the instrumental firmness data and the gene expression data of *CpXTH6* and *CpXTH23*. All the analyses were performed using the InfoStat^®^ software, version 2017.

## 5. Conclusions

Both genes show transcriptional activity in both evaluated stages, but they are more active during the development of the *C. papaya* fruit. The transcriptional activity of the *CpXTH6* and *CpXTH23* genes showed a positive instrumental correlation with firmness during fruit ripening as a result of the decrease in both variables. Therefore, the transcriptional activity of these genes could be related to cell growth and maintenance of firmness of the *C. papaya* fruit, and not to maturity and softening.

## Figures and Tables

**Figure 1 ijms-26-04490-f001:**
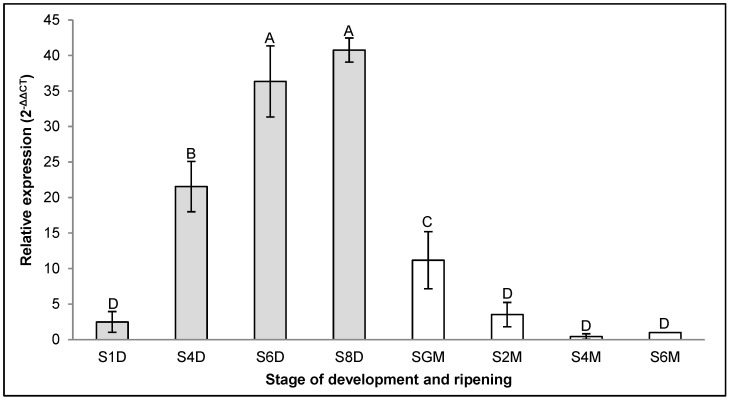
Relative expression of the *CpXTH6* gene in the different development (grey) and ripening (white) stages. The expression level of gene at S6M was used as the control (nominal value = 1). Vertical bars indicate the standard error of three biological replicate assays. Different letters indicate significant difference between the stages (Tukey, *p* < 0.05).

**Figure 2 ijms-26-04490-f002:**
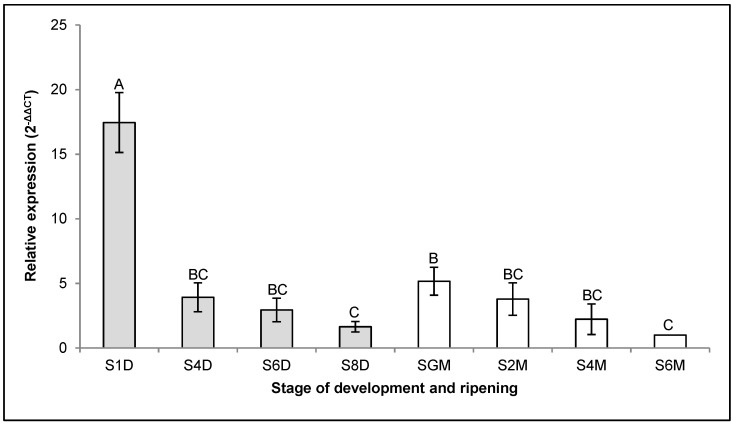
Relative expression of the *CpXTH23* gene in the different development (grey) and ripening (white) stages. The expression level of gene at S6M was used as the control (nominal value = 1). Vertical bars indicate the standard error of three biological replicate assays. Different letters indicate significant difference between the stages (Tukey, *p* < 0.05).

**Figure 3 ijms-26-04490-f003:**
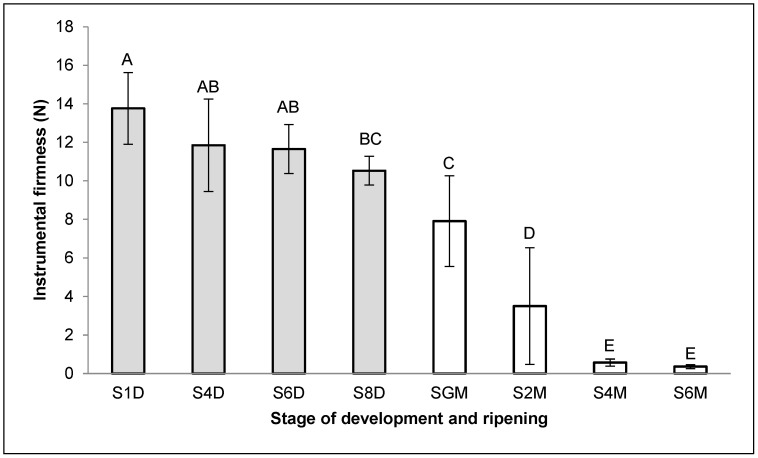
Values of firmness (N) of papaya fruits in the different development (grey) and ripening (white) stages. The bars represent the standard deviation. Different letters indicate significant differences between the stages (Tukey, *p* < 0.05).

**Figure 4 ijms-26-04490-f004:**
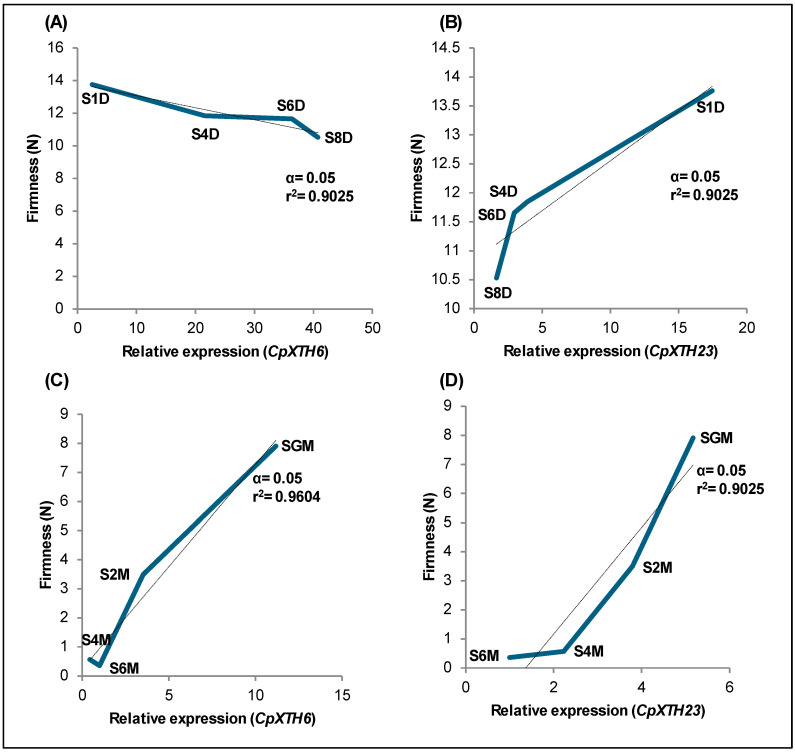
Pearson correlation between the expression of *CpXTH* genes and fruit firmness during papaya fruit’s development and ripening stages. Black lines: the trend of R^2^ values. Blue lines: correlation between relative gene expression and fruit firmness. Correlation of *CpXTH6* (**A**) and *CpXTH23* (**B**) genes at the fruit development stage. Correlation of *CpXTH6* (**C**) and *CpXTH23* (**D**) genes at the ripening stage of the fruit.

**Figure 5 ijms-26-04490-f005:**
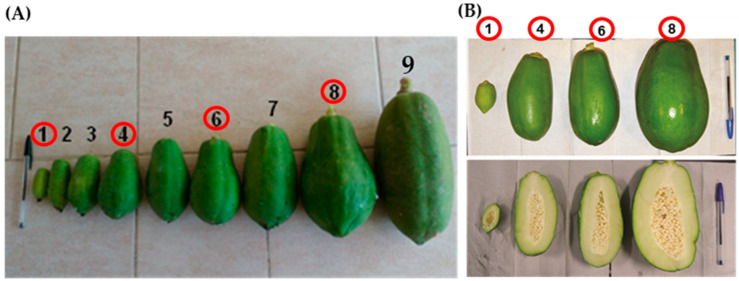
Fruits selected during the development stage of *C. papaya*. (**A**) Scales of complete fruit according to Aguilar-Velázquez et al. (2021) [1]: Stage 1 (SD1: 15 DAA), 4 (SD4: 60 DAA), 6 (SD6: 90 DAA), and 8 (SD8: 110 DAA). (**B**) Internal appearance of the selected fruits. Red circles were the stages analyzed.

**Figure 6 ijms-26-04490-f006:**
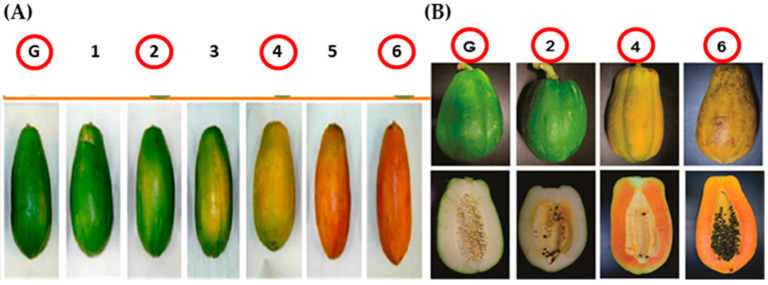
Stages of maturity of papaya fruits. (**A**) Stages of maturity of *C. papaya* according to Santamaría et al. (2009) [23]. (**B**) Stages of ripening of *C. papaya* used for analysis: Stage G (Green; SGM: 0 days of ripeness), Stage 2 (green and yellow stripe; S2M: 6–8 days of ripeness), Stage 4 (orange with some green areas; S4M: 11–13 days of ripeness), and Stage 6 (totally orange; S6M: 15–17 days of ripeness). Red circles were the stages analyzed.

**Table 1 ijms-26-04490-t001:** Applied Biosystems^®^ oligonucleotides and TaqMan Probes used for the qPCR analysis.

Gene	Tm (°C)	F (5′-3′)	R (5′-3′)	TaqMan^®^ Probes
*CpXTH6*	56	TGGACGAAGTTCCGATCAGAGT	CCATGGGTTGGAATTTTGGA	6FAMAAGAACAACGAAGCCAGAAACATCCCGT-QSY
*CpXTH23*	59	TTGATATGCAGCTCAAGCTTGTC	AAGTTGAGAGGCATACATAATAAGCAGTA	6FAMCGGAAACTCCGCTGGCACCG-QSY
Actin	60	TGGTGAAGGCTGGATTTGCT	GTCTAGGACGGCCCACAATACT	VICATGATGCTCCCAGGGCAGTTTTCCC-QSY

## Data Availability

https://drive.google.com/drive/folders/1FOiDk88MrYiB6e7SfPG0F8i_ibIsnHLR?usp=drive_link.

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
