# Peer review of "Expression of the *CpXTH6* and *CpXTH23* Genes in *Carica papaya* Fruits"

_ijms, 2025, doi:10.3390/ijms26104490_

Round 1
Reviewer 1 Report
Comments and Suggestions for Authors
Dear authors,
Thanks a lot for such good work. The manuscript offers an examination of the functions of the CpXTH6 and CpXTH23 genes in the growth and ripening of Carica papaya fruit. Would you please just follow the comments to be able to publish this work.
- The plagiarism percentage needs to be lower than 20%.
- Please mention “papaya” in your keywords after the abstract.
- Would you please clarify more the economic importance of the papaya.
- In table 1: please add cell indicating the Tm of each primer.
- For your comment on figure 3: would you please explain the data in more details.
Author Response
Author's Reply to the Review Report (Reviewer 1)
Comments and Suggestions for Authors:
Thanks a lot for such good work. The manuscript offers an examination of the functions of the CpXTH6 and CpXTH23 genes in the growth and ripening of Carica papaya fruit. Would you please just follow the comments to be able to publish this work.
Reply: Dear reviewer, we thank and appreciate your comments that make us to improve our work. Please, find below the answer for each comment.
- Comments 1: Please mention “papaya” in your keywords after the abstract.
Response: we included already the word “papaya” in keywords. Now it appears:
Texte:
Keywords: Xyloglucan; fruit development; cell wall; qPCR; papaya.
- Comments 2: Would you please clarify more the economic importance of the papaya.
Response: We explained more the economic importance.
Texte:
Mexico is the center of origine of Papaya and many families depends on the pro-duction. In 2023, Mexico produced approximately 1,135,000 tons of papaya, securing fifth place among the world’s top producers of the fruit. Despite not leading in produc-tion volume, Mexico continues to dominate the global export market. With interna-tional sales reaching USD $123.7 million, the country holds the position of the world’s leading exporter of papaya, underscoring its efficiency and competitiveness in interna-tional trade [1].
Comments 3: In table 1: please add cell indicating the Tm of each primer.
Response 3: please, find in red the Tm for each primer.
|
Gene |
Tm (°C) |
F (5’ – 3’) |
R (5’ – 3’) |
TaqMan Probes® |
|
CpXTH6 |
56 |
TGGACGAAGTTCCGATCAGAGT |
CCATGGGTTGGAATTTTGGA |
6FAMAAGAACAACGAAGCCAGAAACATCCCGT-QSY |
|
CpXTH23 |
59 |
TTGATATGCAGCTCAAGCTTGTC |
AAGTTGAGAGGCATACATAATAAGCAGTA |
6FAMCGGAAACTCCGCTGGCACCG-QSY |
|
Actin |
60 |
TGGTGAAGGCTGGATTTGCT |
GTCTAGGACGGCCCACAATACT |
VICATGATGCTCCCAGGGCAGTTTTCCC-QSY |
Table 1. Applied Biosystems® oligonucleotides and TaqMan Probes used for the qPCR analysis.
- Comments 4: For your comment on figure 3: would you please explain the data in more details.
Response 4: please, find in red the explanation.
Text:
The firmness values of papaya fruit during the different stages of development and organoleptic ripeness are shown in Figure 3. The firmness values were statistically similar during the transition from stages S1D to S6D (average 12.8 N). However, in the transition from S1D to S8D, the firmness decreased by 23.54% and was statistically significant (p<0.05). This decline is likely due to S8D being the final stage of development, where fruit softening begins to accelerate. On the other hand, a significant trend was the more accelerated decrease in firmness during the organoleptic ripeness stage, reaching, in stage S6M, values that represented only 3.1% of the average value of stages S1D-S6D (Figure 3), reflecting the rapid breakdown of cell wall components characteristic of this ripening phase [3,4].

Reviewer 2 Report
Comments and Suggestions for Authors
The work of Zúñiga-Hernandez et al. investigates the expression level of two xyloglucan endotransglucosylase/hydrolase (XTH) enzymes, XTH6 and XTH23, in Carica papaya during fruit development and maturation. By quantifying gene expression level, and checking fruit firmness during fruit development and maturation, the authors identified positive correlation between gene expression level and fruit firmness, notably at the ripening stage of the fruit. My main concern is that the analysis relies solely on correlation analysis, I would appreciate the authors to emphasize this in the text and strengthen that further experiments are needed to directly link XTH6 and XTH23 to fruit development and maturation. Apart from that point, the work is clearly described and the manuscript well written, it just requires minor revisions before being suitable for publication in International Journal of Molecular Sciences. Please see below for my major and minor points.
Major points
Line146-147: I agree with the conclusions of the authors. However, as this research is solely based on correlation, I would I suggest adding a note that other XTHs may also be involved in these processes, and that further genetic analysis is needed to fully validate these findings.
Minor points
Figure1-3: Please, indicate in Figure legend what the grey (and white) color corresponds to in the barplot. I understand grey is for fruit development stages, and white for maturation; please indicate it.
Figure4: To avoid any confusion for the readers, please indicate gene name directly in the plot (maybe under brackets after “Relative Expression”? Also, please label the exact developmental stages in panel A and B, as you did for panel C and D.

Author Response
Author's Reply to the Review Report (Reviewer 2)
Comments and Suggestions for Authors:
The work of Zúñiga-Hernandez et al. investigates the expression level of two xyloglucan endotransglucosylase/hydrolase (XTH) enzymes, XTH6 and XTH23, in Carica papaya during fruit development and maturation. By quantifying gene expression level, and checking fruit firmness during fruit development and maturation, the authors identified positive correlation between gene expression level and fruit firmness, notably at the ripening stage of the fruit. My main concern is that the analysis relies solely on correlation analysis, I would appreciate the authors to emphasize this in the text and strengthen that further experiments are needed to directly link XTH6 and XTH23 to fruit development and maturation. Apart from that point, the work is clearly described and the manuscript well written, it just requires minor revisions before being suitable for publication in International Journal of Molecular Sciences. Please see below for my major and minor points.
Reply:
Dear reviewer, we thank and appreciate your comments because it helps to improve our work. Firstly, we are agreeing about your suggestions, is possible that many genes are involved in the process. Please, find added the corrections on the manuscript text in red.
Major points
Line146-147: I agree with the conclusions of the authors. However, as this research is solely based on correlation, I would I suggest adding a note that other XTHs may also be involved in these processes, and that further genetic analysis is needed to fully validate these findings.
Response: Your suggestion was added.
Text:
In addition, the drastic drop in the expression of this gene suggests that a decrease in its expression is required to start the softening of the fruit. However, other XTH members not analyzed may also be involved in these processes, and further genetic analyses are needed to fully validate these findings.
Minor points
Figure1-3: Please, indicate in Figure legend what the grey (and white) color corresponds to in the barplot. I understand grey is for fruit development stages, and white for maturation; please indicate it.
Response: Your suggestion was added.
Text:
Figure 1. Relative expression of the CpXTH6 gene in the different development (grey) and ripening (white) stages. The expression level of gene at S6M was used as the control (nominal value = 1). Vertical bars indicate the standard error of three biological replicate assays. Different letters indicate significant difference between the stages (Tukey, p<0.05).
Figure 2. Relative expression of the CpXTH23 gene in the different development (grey) and ripening (white) stages. The expression level of gene at S6M was used as the control (nominal value = 1). Vertical bars indicate the standard error of three biological replicate assays. Different letters indicate significant difference between the stages (Tukey, p<0.05).
Figure 3. Values of firmness (N) of papaya fruits in the different development (grey) and ripening (white) stages. The bars represent the standard deviation. Different letters indicate significant differences between the stages (Tukey, p<0.05).
Figure4: To avoid any confusion for the readers, please indicate gene name directly in the plot (maybe under brackets after “Relative Expression”? Also, please label the exact developmental stages in panel A and B, as you did for panel C and D.
Response: Your suggestion was added. Please, find the figure with the corrections in the manuscript and the section of figures.
Figure 4. Pearson correlation between the expression of CpXTH genes and fruit firmness during papaya fruit's development and ripening stages. Correlation of CpXTH6 (A) and CpXTH23 (B) genes at the fruit development stage. Correlation of CpXTH6 (C) and CpXTH23 (D) genes at the ripening stage of the fruit.
